# Distinct transcriptomic effects of intermittent and chronic caloric restriction in mammary fat pad of a breast cancer mouse model

Bilge Guvenc Tuna[1]⊕, Nazim Arda Keles[2,3]⊕, Munevver Burcu Cicekdal[4,5], Soner Dogan[2], Sulev Koks[6,7]*

1 Department of Biophysics, School of Medicine, Yeditepe University, Istanbul, Turkiye, 2 Department of Medical Biology, School of Medicine, Yeditepe University, Istanbul, Turkiye, 3 Graduate School of Natural and Applied Sciences, Yeditepe University, Istanbul, Turkiye, 4 Department of Biomedical Molecular Biology, Ghent University, Ghent, Belgium, 5 Department of Biomolecular Medicine, Ghent University and Center for Medical Genetics, Ghent University Hospital, Ghent, Belgium, 6 Perron Institute for Neurological and Translational Science, Nedlands, Washington, Australia, 7 Personalized Medicine Centre, Murdoch University, Perth, Washington, Australia

⊕ These authors contributed equally to this work.
* sulev.koks@murdoch.edu.au

## Abstract

Age-related dysfunction in neuroendocrine signaling, which influences adipose tissue homeostasis, has been implicated in numerous diseases, including breast cancer. Caloric restriction has been shown to improve metabolic health and prolong lifespan, yet the molecular mechanisms underlying its long-term effects are not fully understood. In this study, we investigated the impact of long-term chronic (CCR) and intermittent caloric restriction (ICR) on the whole transcriptome of mammary fat pad tissue (MFP) in a breast cancer mouse model. Transgenic female Mouse Mammary Tumor Virus-Transforming Growth Factor-Alpha (MMTV-TGF-α) C57BL/6 mice were randomized into *ad libitum* (AL), CCR, and ICR groups. Total RNA was isolated from the samples collected at weeks 10 (baseline), 49/50 (adult), and 81/82 (old), were then subjected to RNA sequencing. Differential gene expression analysis identified significant age-related transcriptomic shifts. Specifically, *Malat1* expression levels, a long non-coding RNA associated with cancer progression, were elevated with aging, suggesting increased tumorigenic susceptibility in this model. Pathways linked to neuroendocrine signaling were downregulated with age, reflecting a potential decline in neuro-adipose cross-talk. Remarkably, ICR appeared to mitigate this age-related decline in neuroendocrine signaling by upregulating genes involved in neurotransmitter support and downregulating extracellular matrix organization and positive regulation of angiogenesis. In contrast, CCR did not effectively alter the whole transcriptome profile, particularly in long-term. Our findings reveal that ICR mitigates age-related transcriptional shifts in MFP tissue, providing a novel insight into dietary strategies for maintaining adipose tissue function with potential implications for cancer susceptibility.

**Data availability statement:** All data underlying the findings of this study have been made fully available. The raw sequencing reads have been deposited in the NCBI Sequence Read Archive (SRA) under BioProject ID PRJNA1279411.

**Funding:** This work was supported by Perron Institute for Neurological and Translational Science, Nedlands, WA, Australia. This work was partly supported by The Scientific and Technological Research Council of Turkiye (TUBITAK). The funders had no role in study design, data collection and analysis, decision to publish, or preparation of the manuscript.

**Competing interests:** The authors have declared that no competing interests exist.

## 1. Introduction

Aging leads to inevitable changes in biological processes, including significant transcriptomic modifications at the molecular level [1,2]. These age-related molecular alterations are associated with numerous diseases, including breast cancer [3–5]. Caloric restriction (CR), one of the proposed preventive strategies against mammary tumor (MT) development, is shown to lower breast cancer risk, especially in genetically predisposed individuals [6,7]. CR is defined as reducing caloric intake without causing malnutrition [8] and emerged as a non-pharmaceutical intervention that improves metabolic health by modulating inflammation, oxidative stress, and cellular aging [9–11]. Alternative strategies, such as intermittent CR (ICR) have been implemented by fasting or limiting energy intake during certain periods and could improve metabolic flexibility and cellular stress responses [12,13]. While the effects of chronic CR (CCR) have been extensively studied, the long-term molecular impacts of ICR remain relatively elusive. Therefore, investigating tissue-specific transcriptomic modifications of cancer predisposition models under different CR protocols during aging holds great potential for developing preventive approaches against age-related metabolic diseases and cancer.

Mammary fat pad (MFP), a heterogenous tissue central to metabolic homeostasis, is a key site for investigating the transcriptomic impacts of aging and CR due to its role in both metabolism and cancer, including breast cancer [14,15]. Previous studies reported age-related increase in serum pro-inflammatory cytokines [16], and elevated levels of inflammation in adipose tissue [17]. Although the exact molecular mechanisms remain to be elucidated, previous work showed CR alters the transcriptomic profile and the age-related molecular changes in adipose tissue function [18–20]. Therefore, CR offers a promising strategy to prevent age-related changes in adipose tissue, particularly in individuals with genetic predisposition to breast cancer. Our previous work demonstrated that ICR had protective effects against mammary tumor (MT) development and reduced the tumor incidence rate in the breast cancer mouse model to 11.5%, while CCR and AL group showing 20% and 45.5% incidance rates, respectively. [21]. In contrast, a recent study revealed that a different protocol of ICR was less effective in reducing mammary tumor incidence (20.5%) compared to AL-fed mice (21.1%), while CCR still showed a substantial reduction to 8.8% [22]. These contradictory results highlight the need to investigate the molecular background of different CR strategies, particularly in long term.

In the current study, we hypothesized the long-term CCR and ICR would differentially modulate the transcriptomic landscape of MFP tissue, with ICR potentially exerting more pronounced effects due to the metabolic shifts. Using the MMTV-TGF-α C57BL/6 female mouse model, we aimed to evaluate transcriptomic alterations induced by aging and different CR strategies. The mouse models are foremost for studying long-term dietary interventions and aging since it is challenging for human subjects, with different physiological conditions to comply with the experimental protocols [23]. Thus, we used MMTV-TGF-α breast cancer mouse model which overexpress human TGF-α, a part of the epidermal growth factor receptor (EGFR)/Erb cascade that play a key role in MT development. These mice usually develops MTs in the second half of their lives [24].

Here, the effects of CCR and ICR on the MFP tissue transcriptome and its biological relevance were investigated by analyzing and comparing the gene expression profiles with the AL-fed control mice. Furthermore, we aimed to explore whether the re-feeding period of ICR influences the transcriptional landscape of MFP tissue, following the restriction period. We reported an age-related decline in neuroendocrine signaling in MFP tissue. However, these alterations were reverted with the ICR application. In addition, ICR has a more pronounced effect on MFP tissue transcriptome in the long-term, compared to CCR. Overall, our study provides novel insights into the molecular mechanisms underlying the distinct effects of CCR and ICR on MFP tissue biology and its relationship with aging processes, with potential implications for breast cancer susceptibility.

## 2. Materials and methods

### 2.1. Animals and experimental design

MMTV-TGF-α C57BL/6 female mice that did not develop any mammary tumors (MTs) were used in the current study. Originally, MMTV-TGF-α gene-positive [24] female mice were provided by Dr. Margot Cleary, Hormel Institute, University of Minnesota to establish a breeding colony at Yeditepe University, Istanbul. A total of 46 mice were enrolled in the study (n = 4–5) (S1 Table). All mice were fed AL until 10 weeks of age and then randomly assigned to three different dietary groups, as explained in detail in our previous studies [13,22]: *Ad libitum* (AL), chronic CR (CCR), or ICR (Fig 1A). All the animals were fed with standard chow (Altromin TPF1414) diet purchased from Kobay AS (Ankara, Turkiye). Dietary composition is given as supplementary data (S1 File). Mice in the AL group had free access to food throughout the study. Mice assigned to the CCR group were provided 85% of the daily food consumption (15% CR) of the age-matched AL group, resulting in a 15% caloric reduction compared to the AL group. Mice assigned to the ICR group were provided 40% of the daily food consumption (60% CR) of the age-matched AL group for one week, then were fed AL for the following three weeks, cyclically. This also resulted in an overall caloric reduction of 15% in four weeks compared to age-matched AL mice. Mice in the ICR group were later divided into two groups, according to their sacrification points. The ones that were sacrificed at the end of three weeks of AL feeding (weeks 49 and 81) were referred to as ICR-refed (ICR-RF), while the ones that were sacrificed at the end of the one-week of the CR period (weeks 50 and 82) were referred to as ICR-restricted (ICR-R) (Fig 1A). Body weights and food consumption were measured weekly and reported in our previous publications [13,22]. All mice were fasted overnight and sacrificed the next day at designated time points: weeks 10 (baseline), 49/50 (adult), and 81/82 (old). Animals were anesthetized using isoflurane and humanely euthanized by isoflurane overdose under observation to eliminate unexpected suffering. and MFP tissue samples were collected and stored at −80°C freezer until further use. All efforts were made to minimize suffering, including regular monitoring of health status and the use of humane endpoints when necessary. The health status of the animals was checked weekly by a veterinarian. Mice were allowed free access to tap water and were housed under standard conditions in a room at a temperature of 21–24°C 12h light/dark cycle. All procedures involving animals were in compliance with the European Community Council Directive of 24 November 1986, and ethical approval was granted by the Yeditepe University Ethics Committee (file no. 11 03 2019, Istanbul, Turkiye).

### 2.2. Tissue collection and histopathological analysis

At sacrifice, MFP tissue samples were removed and checked to assess whether there were any macroscopic pathological changes. A piece of each tissue sample was placed in 10% neutral buffered formalin to be sent to the pathologist for histopathologic analyses to determine malignancy and disease status in a blind fashion by the pathologist. The remaining tissues were stored at −80°C until used. The animals included in the current study were tumor-free and healthy animals.

### 2.3. Total RNA isolation and whole transcriptome sequencing

Total RNA was isolated from the frozen MFP tissues stored at -80°C. Tissue samples were homogenized in Direct-zol (Zymo Research, Irvine, CA, USA) with a tissue homogenizer. After homogenization, the remaining tissue

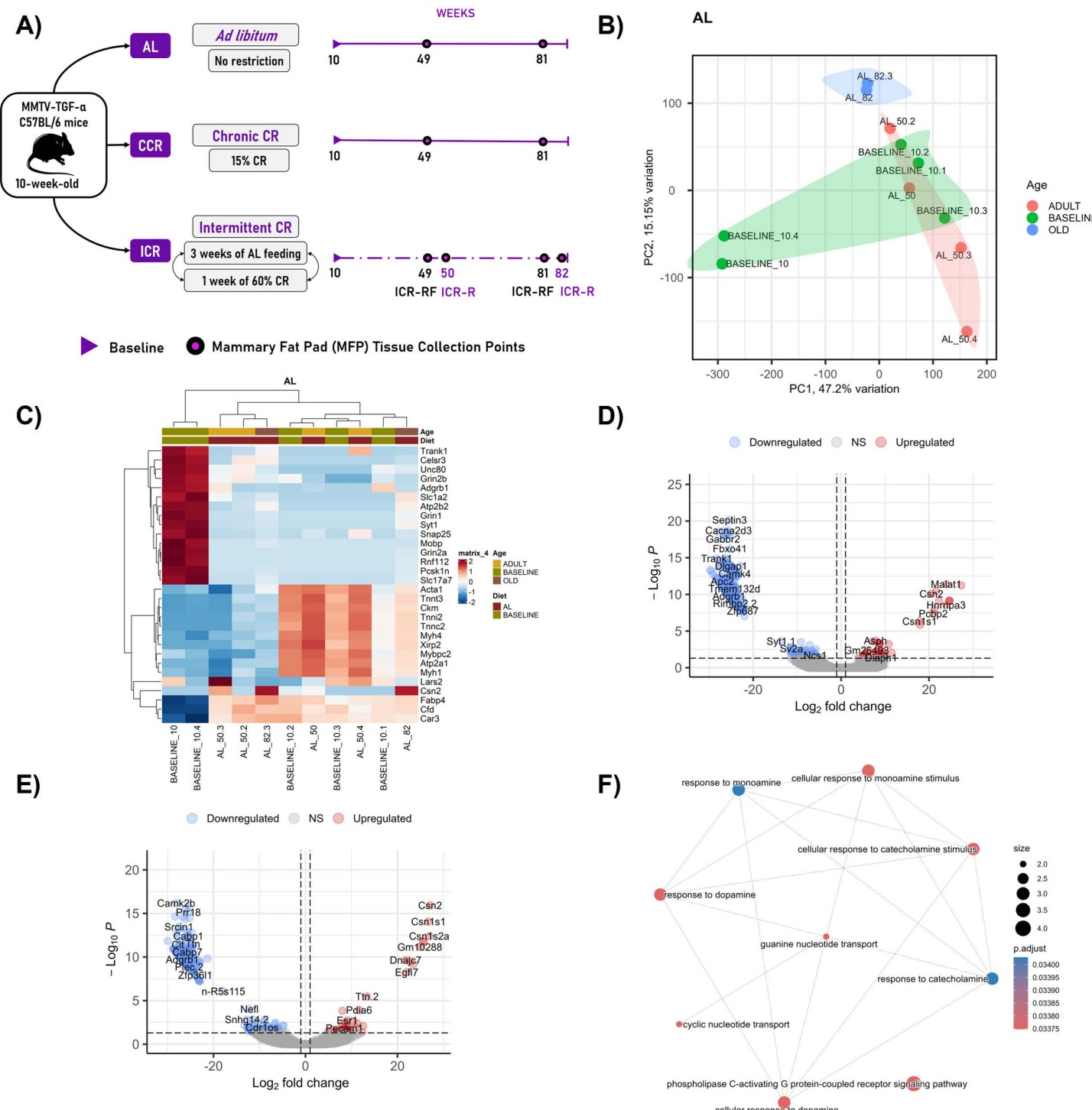

**Fig 1. Age-related modifications in the whole transcriptome and biological pathways of the MFP tissue of the breast cancer mouse model.**
**A)** Experimental design of the dietary groups. AL group followed an *ad libitum* diet (no restriction) throughout the study. CCR group was applied to 15% caloric restriction throughout the study. ICR group was applied to 3 weeks of *ad libitum* feeding followed by 1 week of 60% caloric restriction, cyclically. Samples were collected at weeks 10 (baseline), 49 or 50 (adult), and 81 or 82 (old). AL: *ad libitum*, CCR: chronic caloric restriction, ICR: intermittent caloric restriction, ICR-R: intermittent caloric restriction-restricted, ICR-RF: intermittent caloric restriction-refed.**B)** PCA analysis of the samples belong to AL group at different time points. **C)** Heatmap displays the unsupervised hierarchical clustering of the expression patterns of the top 30 most variable

genes in the AL group at different time points. **D-E)** Volcano graphs display the differentially expressed genes (DEGs) compared to the baseline levels at adult and old age, respectively. **F)** Network plot displays the relationship between the enriched pathways of downregulated genes with adult AL compared to the baseline, with adjusted p-values. PC: Principal component. For volcano graphs, absolute log2 fold-change value of > 1 and adjusted p value of 0.1 were considered statistically significant.

residual was removed from suspension and RNA isolation was performed with a Direct-zol RNA MiniPrep Kit according to the manufacturer's instructions (Zymo Research, Irvine, CA, USA, Cat No: R2052). RNA concentrations were determined with a NanoDrop 2000 spectrophotometer (ThermoFisher Scientific, Waltham, MA, USA) and total RNA integrity was checked by 1% agarose gel electrophoresis. RNA samples were sequenced using the Illumina Novaseq (Illumina, San Diego, CA, USA) sequencer at the Murdoch University Genomics Centre, Perth, WA, Australia.

### 2.4. Bioinformatic and statistical analyses

The resulting *fastq* files were used for whole transcriptome analysis using Salmon workflow as described in detail previously [25]. In brief, the index and decoy were based on mouse genome assembly version GRCm39, Gencode release M29. Salmon performs quasi-mapping and provides highly accurate expression measures as *quant* files. The quant files were imported to R using the *tximport* function with gene-level summarisation for further analyses. Differential expression analysis was done in different stages. First, the effect of aging for each dietary group was analyzed, and then the different diets at each time point were compared to each other (at weeks 49/50 and 81/82). The baseline for analysis was at week 10 when all the animals finished AL feeding and switched to the specific experimentally allocated feeding schemes. R programming language (version 4.3.3) was used for all the bioinformatic and statistical analyses and visualizations [26]. *DESeq2* package with standard workflow was used for differential expression analysis [27]. Statistical significance testing is done with *vegan* package using the PERMANOVA test. *PCAtools* package was used to visualize principal component analysis (PCA) results. For the enriched pathway analysis and visualizations, *clusterProfiler and enrichplot* packages were used. For the reference pathway annotations, *org.Mm.e.g.,db* package was used. FDR correction was applied for the multiple testing of whole transcriptome data using the Benjamini-Hochberg method [28]. The significance threshold was set at a p-value of < 0.05, FDR-adjusted p-value of 0.1, and absolute log2 fold-change (log2FC) of > 1.

### 3. Results

#### 3.1. Age-related alterations in the whole transcriptome profile of mammary fat pad tissue

All the tissue samples used in the current study were collected from healthy mice that did not develop any MTs. RNA-sequencing of MFP tissue samples from mice generated an average of 2,222,240 raw reads per sample. After quality filtering, an average of 2,217,814 high-quality reads per sample remained. On average, 11,579 genes were detected per sample, with the number of expressed genes ranging from 315 to 16,378 across all samples. RNA-sequencing results of adult (49/50-week-old) and old (81/82-week-old) mice in the AL group were compared with the baseline (10-week-old) to investigate the age-related changes in the MFP transcriptome. PCA revealed that aging did not significantly distinguish the global transcriptome profile (p > 0.05) (Fig 1B). A few individual samples had unique transcriptome profiles, which might indicate inter-individual variability or localized transcriptional alterations. Although PCA showed no significant global distinction in whole transcriptome profiles, differential expression analysis highlighted specific gene-level alterations with age. The heatmap of unsupervised hierarchical clustering displays the expression patterns of the top 30 most variable genes between the different time points in the AL group (Fig 1C).

Differential gene expression analyses revealed age-related alterations in the AL-fed mice (Fig 1D,E). In adult mice, the number of differentially expressed genes (DEGs) compared to the baseline was 668 (77↑, 591↓), while in old mice, there were 466 DEGs compared to the baseline levels (44↑, 422↓). Among the upregulated genes in adult mice, the *Asph* gene is thought to play an important role in calcium homeostasis. Additionally, *Hnrpna3* and *Pcbp2* are predicted to be involved in RNA binding, transcription, and splicing activity. Interestingly, *Malat1* is also upregulated in adult mice compared to baseline. *Malat1* gene product is a long non-coding RNA that functions as a gene expression regulator and is linked to various types of cancer. Upregulation of *Malat1* might suggest age-related susceptibility to increase in oncogenic pathways in MFP. *Csn1s1* and *Csn2* was consistently upregulated with age compared to the baseline levels. Not surprisingly, *Csn* beta-casein gene family is highly expressed in the mammary gland. This suggests its sustained involvement in mammary gland biology, potentially beyond its lactation-related roles. Most downregulated genes such as *Septin3*, *Cacna2d3*, and *Fbxo41* in adult mice are known to be involved in neuronal communication and presynapse organization. These genes were mostly enriched in cellular response to dopamine, catecholamine, and monoamine pathways, which points to potential alterations in neuroendocrine regulation, possibly reflecting age-related dysfunction (Fig 1D). In old mice, in addition to the beta-casein gene family, *Dnajc7*, *Egfl7*, *Pcbp2*, and *Pdia6* were among the upregulated genes compared to the baseline. *Pdia6* enables protein-disulfide reductase activity and is involved in unfolded protein response. *Pcbp2* is involved in the positive regulation of transcription by RNA polymerase II. On the other hand, *Egfl7* is involved in the negative regulation of smooth muscle cell migration. *Dnajc7* is predicted to be involved in heat shock protein binding activity. Among the downregulated genes, *Camk2b* enables calcium/calmodulin-dependent protein kinase activity and is involved in the regulation of neuron migration. *Srcin1* enables protein kinase binding activity and involved in actin cytoskeleton organization.

Overall, these results suggest that aging is a major regulator of gene expression related to neuroendocrine signaling in the MFP tissue of the transgenic MMTV-TGF-α breast cancer mouse model, potentially shaping a microenvironment supports the tumor initiation and progression in this model.

## 3.2. Mid-term effects of chronic or intermittent caloric restriction on mammary fat pad tissue transcriptome in adult mice

RNA sequencing results of the CCR, ICR-R, and ICR-RF groups were compared to the AL group in adult mice to investigate the mid-term effects of different dietary interventions (chronic or intermittent CR) on MFP transcriptome. PCA showed no significant differences in the global transcriptome profiles of MFP tissue between the groups in adult mice ($p > 0.05$) (Fig 2A). Although PCA did not reveal significant global distinction in whole transcriptome profiles, differential expression analysis identified specific gene-level changes associated with diet, suggesting localized transcriptomic shifts. The heatmap of unsupervised hierarchical clustering displays the expression patterns of the top 30 most variable genes between the different dietary groups in adult mice (Fig 2B).

Differential expression analyses revealed specific DEGs between the dietary groups in adult mice. In the CCR group, there were 10 DEGs compared to the AL group (5↑, 5↓). Among the upregulated genes, *Stxbp1* and *Snph* has a role in membrane fusion and vesicle organization (Fig 2C). In contrast to CCR, the ICR-R demonstrated a more pronounced impact on the transcriptomic profile of MFP tissue, with 637 DEGs identified compared to the AL group (455↑, 182↓) (Fig 2D). In the ICR-R group, upregulated genes include *Slc17a*, an L-glutamate transmembrane transporter, and *Pcsk1n*, which enables serine-type endopeptidase inhibitor activity and is involved in peptide hormone processing, response to cold, and response to dietary excess. Upregulated genes in the ICR-R group were mostly enriched in the vesicle-mediated transport in synapse, cognition, and neurotransmitter transport, indicating potential neuroprotective effects (Fig 2E). Among the downregulated genes, *Col1a1* and *Col1a2* encode collagen, which is primarily found in connective tissues. The *Tert* gene encodes the telomerase enzyme, whose activity is associated with angiogenesis. Downregulated

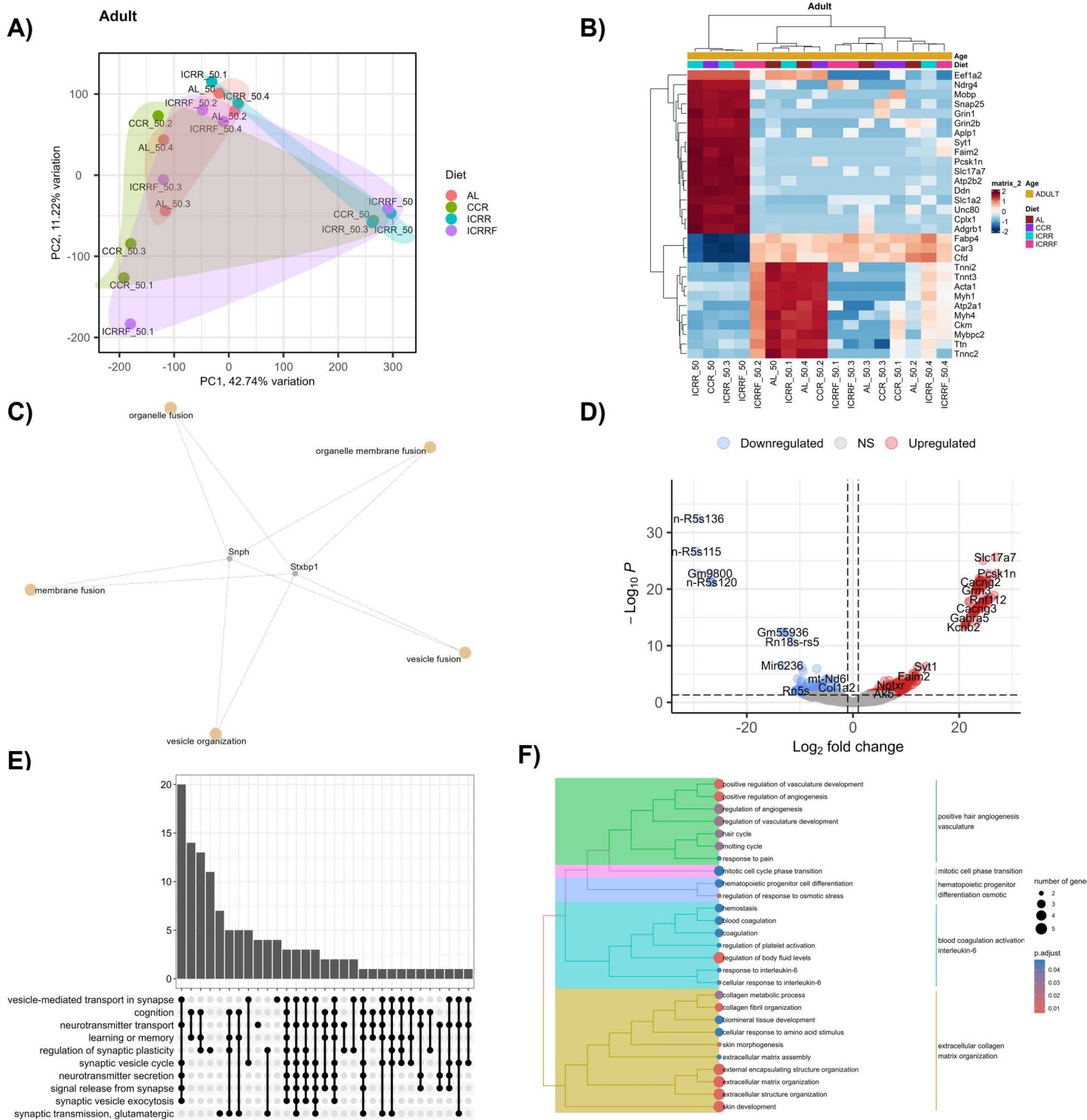

**Fig 2. Mid-term effects of the different dietary interventions in the MFP tissue transcriptome and the biological pathways of the breast cancer mouse model in the adult age. A)** PCA analysis of the samples belong to different dietary groups at adult age. **B)** Heatmap displays the unsupervised hierarchical clustering of the expression patterns of the top 30 most variable genes in different dietary groups at adult age. **C)** Network plot displays the relationship between the upregulated genes in the CCR group and the enriched pathways. **D)** Volcano graph displays the DEGs between ICR-R and

AL group, at adult age. **E)** Upset plot represents the enriched pathways of the DEGs in the ICR-R group in adult mice, with the common gene counts. **F)** Tree plot represents the enriched pathways of downregulated genes in the ICR-R group in adult mice. PC: Principal component. For volcano graphs, absolute log2 fold-change value of > 1 and adjusted p value of 0.1 were considered statistically significant.

genes in ICR-R, compared to the AL group, were mostly enriched in extracellular matrix organization and positive regulation of angiogenesis, suggesting potential downregulation in tissue remodeling and pathways related to the cancer development in the MFP microenvironment (Fig 2F). In the ICR-RF group, there were 73 DEGs compared to the AL group in adult mice (3↑, 70↓), indicating a three-week re-feeding period resulting in the loss of significant differences in a vast number of genes (S1 Fig).

Overall, these results suggest the ICR application was much more effective than the CCR application on MFP tissue whole transcriptome in adult mice, considering the number of DEGs in the groups. On the other hand, ICR application mostly upregulated the pathways related to neurotransmitter support and regulation of synaptic plasticity, with potential neuroprotective effects, while downregulating the extracellular matrix organization and angiogenesis, a possible tumor-supporting microenvironment.

### 3.3. Long-term effects of chronic or intermittent caloric restriction on mammary fat pad tissue transcriptome in old mice

The transcriptomic profile of MFP tissue of the CCR, ICR-R, and ICR-RF groups was compared to that of the AL group old mice to investigate the long-term effects of CR on the MFP tissue transcriptome. PCA showed no significant global transcriptomic distinctions among dietary groups in old mice (p > 0.05), though individual variability suggested localized transcriptional alterations (Fig 3A). Although PCA showed no significant global distinction in whole transcriptome profiles, differential expression analysis highlighted specific gene-level alterations with diet. The heatmap of unsupervised hierarchical clustering displays the top 30 most variable genes between the dietary groups in old mice (Fig 3B).

Differential expression analysis revealed distinct responses to different dietary interventions in old mice. Notably, there were no DEGs were identified in the CCR group compared to AL-fed old mice suggesting that continuous application of CR, when sustained long-term, did not effectively counteract the age-related changes in MFP tissue transcriptome. On the other hand, long-term ICR-R application resulted in 90 DEGs (57↑, 33↓) compared to AL feeding (Fig 3C). Among the upregulated genes, *Cpt1b* is a member of carnitine/choline acetyltransferase family which rate-controls long-chain fatty acid beta-oxidation pathway in muscle mitochondria, which indicates a possible metabolic adaptation response to ICR in the long-term. *Trim63* is a RING zinc-finger protein family which mostly found in striated muscles. In the ICR-R group, upregulated genes were predominantly enriched in pathways related to muscle development and differentiation, indicating a potential diet-induced enhancement of muscle-related processes, which may counteract age-associated decline (Fig 3D, S2 Fig). Enriched terms in the ICR-R group in old mice revealed that ICR-R strongly modified muscle development since most of the enriched terms are intensely connected (Fig 3E). While most of the genes related to these enriched terms were upregulated, down-regulated genes including *Lep*, which is an appetite regulator and has a role in cytokine signaling and metabolism, and *Atg7* is involved in autophagy and modulates p53-dependent cell cycle signaling. (Fig 3F). Among the downregulated genes, *Nnat* has a role in the positive regulation of insulin secretion and response to glucose. *Sez6l* gene is predicted to be involved in protein kinase-C signaling and synapse maturation (S3 Fig).Interestingly, there were no DEGs in the ICR-RF group compared to the AL group in old mice. The absence of DEGs in the ICR-RF group suggests that re-feeding reverted the transcriptomic modifications induced by restriction, potentially reflecting a loss of CR's beneficial effects (S4 Fig).

These findings suggest that ICR application, when applied in the long term up to 82 weeks of mouse age, could modify the age-related transcriptomic changes in the MFP tissue of the breast cancer mouse model. On the other hand, continuous application of more moderate restriction (CCR) was not effective in transcriptomic regulation since there were no

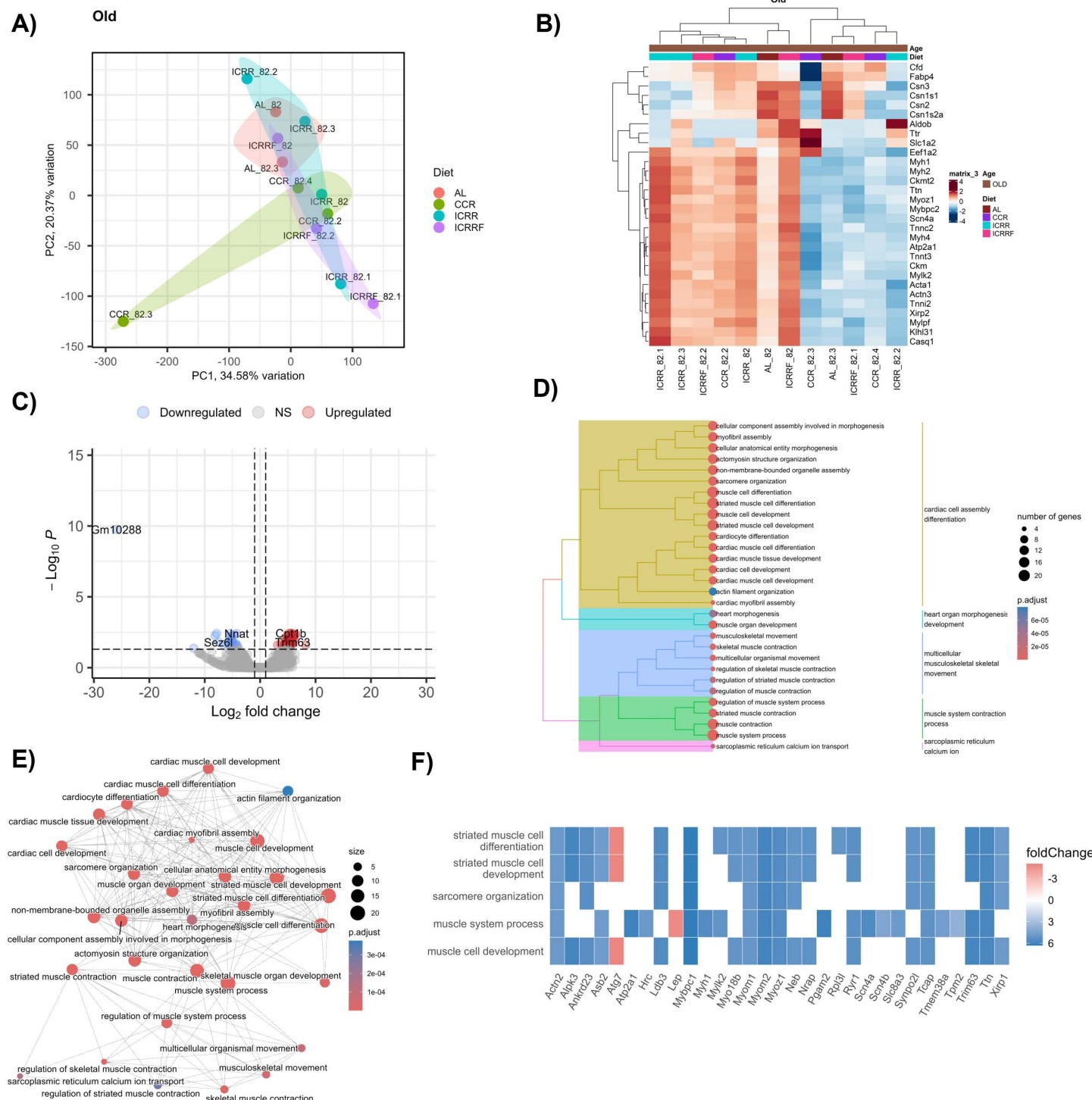

**Fig 3. Long-term effects of the different dietary interventions in the MFP tissue transcriptome and the biological pathways of the breast cancer mouse model at old age. A)** PCA analysis of the samples belong to different dietary groups at old age. **B)** Heatmap displays the unsupervised hierarchical clustering of the expression patterns of the top 30 most variable genes in different dietary groups at old age. **C)** Volcano graph displays the DEGs in the ICR-R group compared to AL group at old age. **D)** Tree plot represents the hierarchical clustering of the enriched pathways of the DEGs

in the ICR-R group. **E)** Network plot display the connection of genes with the enriched pathways in ICR-R group at old age. PC: Principal component. For volcano graphs, absolute log2 fold-change value of >2 and adjusted p value of 0.1 were considered statistically significant. **F)** Heat plot displays the relationship between the DEGs and the enriched pathways with fold-change values.

DEGs in the CCR group compared to AL-fed old mice. DEGs in the ICR-R group were mostly involved in muscle development, indicating the diet-induced regulations in the muscle system process.

## 4. Discussion

In the current study, we performed RNA sequencing on MFP tissue of the transgenic breast cancer mouse model subjected to different dietary regimens (AL, CCR, or ICR) from week 10 up to 82 to investigate transcriptomic alterations. Differential expression analysis revealed significant age-associated shifts, including a downregulation in neuronal communication, presynapse organization, cellular response to dopamine, monoamine, and cotacholamine, which suggests possible neuroendocrine dysfunction. In this context, Willows et al. demonstrated that subcutaneous adipose tissue contains myelinated and unmyelinated nerves and is densely populated with Schwann cells associated with synaptic vesicle-containing nerve terminals [29]. In addition, a previous study demonstrated that hypothalamic-adipose interconnection regulates aging and lifespan in mice [30]. Consistent with our findings, prior studies have reported an age-related decline in neuroendocrine signaling [31,32]. This suggests a potential systemic mechanism that impacts not only neuronal tissues but also distinct environments like MFP. The consistent downregulation of synaptic function-related genes in MFP tissue could reflect an age-associated reduction in neuroendocrine signaling, potentially contributing to impaired metabolic regulation or local tissue remodeling.

In the present study, we reported a consistent age-related increase in *Csn1s1* and *Csn2* expression in AL-fed mice. While these genes primarily function in lactation, a previous study using TCGA breast cancer datasets reported lower *CSN1S1* expression in breast cancer tissues compared to healthy breast tissue, and this lower expression is associated with better prognosis [33]. On the other hand, Peng et al. found that *CSN1S1* was upregulated in breast cancer patients with distant metastases, compared to those without [34]. However, the molecular evidence remains limited regarding how these genes contribute to breast cancer initiation and progression, highlighting the need for further investigation into both *CSN1S1 and CSN2* as potential prognostic markers.

Notably, an age-related increase in *Malat1* expression level was observed in adult mice, while no such change was detected in old mice. *Malat1* gene product is a long non-coding RNA that regulates gene expression and is linked to various types of cancer [35,36]. Xiping et al. reported that *MALAT1* expression was positively correlated with metastatic lymph nodes in breast cancer patients and promotes proliferation and invasion abilities through XPB1-HIF −1α and Her2 pathways in MDA-MB-231 and MDA-MD-435 breast cancer cell lines, respectively [37]. In contrast to our findings, previous studies reported reduced *MALAT1* expression with age in skeletal muscle and adipose tissue [38,39]. However, Arshi et al. reported consistent and age-independent upregulation of *MALAT1* levels in women with breast cancer [40]. In the present study, we did not observe any change in *Malat1* expression with CR, which might indicate *Malat1*-associated pathways are less responsive systemic modulations in the metabolism of this model. This could reflect a possible susceptibility to oncogenic molecular changes despite a CR protocol in a breast cancer mouse model, supported by the previous findings [41].

After demonstrating the age-related transcriptomic alterations in MFP tissue, we next explored how dietary interventions might counteract these changes. CCR has been extensively studied for its beneficial effects on longevity and metabolic health and has shown varying impacts in different tissues [42]. Here, we assessed whether mid- and long-term CCR could mitigate the age-associated transcriptomic shifts observed in MFP tissue. The findings revealed that CCR limitedly affected gene expression when applied to the mid-term, contrasting with previous findings [43,44]. *Stxbp1* and *Snph* were among the upregulated genes with CCR, which are involved in key signaling pathways related to release of

 

neurotransmitters, synaptic vesicle docking, and mitochondrial trafficking. In humans, *SNPH* encodes a mitochondria anchoring protein negatively regulates mitochondrial trafficking in neurons. *SNPH* has been reported to be downregulated among distinct tumor types, including breast tumors, and its loss is associated with tumor progression, metastasis, and poor prognosis of the patients [45,46]. In this context, the upregulation of *Snph* with CCR may suggest a potential protective effect against the breast cancer initiation and metastasis through altered mitochondrial trafficking in the MFP. In addition, *STXBP1* is a regulator of lysosome-dependent cell death and has been shown to be downregulated in HR +/ HER- breast cancer, compared to adjacent healthy tissues. Although *STXBP1* downregulation has been associated with better prognosis, the precise role of *STXBP1* gene in breast remains unclear and requires further investigation [47]. Overall, the limited number of DEGs in observed with mid-term CCR suggests a subtle impact on the MFP transcriptome that appears insufficient to counteract the broader age-related transcriptomic alterations.

Interestingly, long-term CCR-applied mice showed no significant alterations in the whole transcriptome, compared to AL-fed mice. This could be attributed to its less intense restriction amount. Unlike ICR, which includes refeeding cycles that may enhance metabolic flexibility and stress adaptation, CCR imposes a continuous but moderate caloric deficit. For instance, a previous study reported the reduced impact of CR over prolonged periods [48], while Higami et al. reported 9-month of CR was much more effective than 23 days of CR 10-month-old mice [49]. These findings emphasize the importance of dietary intervention strategies in modulating age-related transcriptomic changes and suggest that the intensity and pattern of CR play crucial roles in determining their effectiveness.

Intermittent fasting, or ICR, was previously reported to regulate gene expression [50], ameliorate adipose tissue-associated inflammation [51], and promote energy expenditure [52]. In the present study, long-term ICR significantly altered MFP transcriptome compared to AL-feeding, more effectively than CCR. ICR-R upregulated genes, such as *Pcsk1n*, and *Slc17a7*, which are involved in transmembrane transport, membrane potential, synaptic plasticity, and neurotransmitter support, pointing to potential neuroprotective and metabolic effects. *PCSK1N* regulates neuropeptide processing essential for energy homeostasis and appetite regulation [53], and its expression has been linked tobody mass index (BMI), age, and type 2 diabetes [54–56], while contradictory findings were also reported [53]. *SLC17A7* encodes VGLUT1, which modulates neuronal communication and synaptic plasticity [57]. Its expression in the mouse hypothalamus was upregulated with a high-fat diet [58], while it was downregulated with a ketogenic diet [59] indicating the potential regulatory role of *Slc17a7* in different dietary interventions. These signaling pathways related to membrane potential and synaptic communication were downregulated with aging. However, ICR appeared to counteracts these age-related alterations, suggesting a potential role in mitigating age-related neuroendocrine dysfunction associated with aging. On the other hand, downregulated genes in ICR-R, compared to the AL group, were mostly enriched in extracellular matrix organization and positive regulation of angiogenesis, suggesting potential downregulation in tissue remodeling and pathways related to the cancer development in the MFP microenvironment [60,61].

Notably, we reported reduced leptin (*Lep*) expression with ICR in old mice, consistent with the previous findings [49,62]. Leptin and adiponectin are key adipokines secreted from adipose tissue that regulate energy homeostasis and metabolic response, with an abnormal adiponectin/leptin ratio linked to diseases including obesity [63,64]. Earlier studies also reported increased adiponectin/leptin ratio in serum and plasma with CR [65,66] and their role in breast cancer incidence [21]. In this context, changes in leptin or adiponectin signaling-related miRNAs have also been described [22]. Leptin exerts its biological functions through selective binding to its receptor (LEPR), which is expressed in many tissues including the mammary glands [67]. Prior studies demonstrated that leptin stimulates the growth of human breast cancer cells *in vitro* and promotes angiogenesis via vascular endothelial growth factor (VEGF) and VEGF receptor 2 (VEGF2) pathways [68].

Beyond their involvement in breast cancer progression, leptin and adiponectin plays a central role in metabolism, regulating appetite, energy balance, and adipose tissue function. A previous study in Wistar rats reported that just two days of refeeding upregulated the *Lep* levels and reversed the CR effect [69]. Similarly, Liu et al. reported a 32.8% reduction in the *Adipoq* expression, after the two-day refeeding period [70]. In line with these findings, our study showed that a three-week

re-feeding period in ICR reversed many transcriptomic differences and resulted in a marked reduction in DEGs. This may reflect an adaptive response to short-term caloric intake variations, consistent with prior reports [69,71]. Therefore, dietary interventions like ICR could be optimized to balance benefits with minimal adverse effects from refeeding.

We previously reported a significant decrease in MT development in mice exposed to ICR for up to 74 weeks compared to the AL-fed mice [21]. However, the exact molecular mechanisms of the protective effects of ICR are still unclear, although several molecular pathways have been associated with this phenomenon, including oxidative stress [13]. Therefore, the findings in the present study indicate that ICR may have a protective role by globally modulating the MFP tissue gene expression profile against age-related alterations.

Nevertheless, our study has a few limitations. For example, the functional reflections of transcriptomic alterations are yet to be clarified, therefore it is important to examine the effects of CR on gene expression in MFP tissues by experimentally validating the findings, including qPCR, western blot, or immunohistochemistry techniques. In future studies, more omics studies are needed for an integrative approach. In addition, the absence of global transcriptomic distinctions in PCA, despite the significant number of DEGs might reflect sample size limitation or variability. Moreover, we did not include wild-type mice in the current study, since all mice have MMTV-TGF-α genetic background. Previous studies demonstrated that the genetic background of the animals could affect the metabolic response to CR [72,73].

We acknowledge that the use of a single transgenic breast cancer mouse model used in this study, MMTV-TGF-α, may not fully capture the heterogeneity of breast cancer subtypes. This model was chosen to investigate the long-term effects of CR against MT development, as it develops MTs with a long latency (6−15 months) [24]. However, other widely used models, such as FVB/N-Tg(MMTVneu)202Mul/J and FVB/N-Tg(MMTV-Erbb2*,-cre)1Mul/J [74,75] do not require multiple pregnancies for rapid tumor development. Notably, a prior study applying the same ICR protocol to MMTV-neu mice -previously applied to MMTV-TGF-α mice- reported tumor incidences 37% in AL-fed mice and 22% in ICR mice [76], suggesting that ICR might be effective in reducing the MT incidence regardless of the model. Given the hormone-independent and aggressive nature of MMTV-neu and Erbb-2 driven tumors, compared to MMTV-TGF-α mice used in the current study, it is possible that CR interventions may exert distinct molecular impact. Therefore, while the protective effects of CR, particularly ICR, appear promising, further comparative studies including multiple transgenic breast cancer models are needed to determine how ICR influences the tissue transcriptome and MT development across diverse genetic backgrounds.

## 5. Conclusion

In conclusion, we demonstrated for the first time the long-term application of ICR, but not CCR, substantially alters the global transcriptome of MFP tissue of the transgenic breast cancer mouse model. While ICR holds potential as a non-pharmaceutical prevention strategy against age-related transcriptomic alterations, further research is needed to comprehend the underlying epigenetic mechanisms. Identifying molecular signatures related to ICR-mediated gene expression could lead to the development of personalized dietary interventions targeting age-related metabolic and transcriptional dysregulation.

## Supporting information

**S1 File. Dietary composition of the food given to the mice throughout the study (Altromin TPF1414).**
(PDF)

**S1 Table. Number of the MFP tissue samples for each dietary group at each time point of the study.**
(DOCX)

**S1 Fig. Genes affected by refeeding at adult age.** Volcano graph displays the differentially expressed genes (DEGs) in the ICR-RF group compared to the ICR-R group at adult age. Absolute log2 fold-change value of >2 and adjusted p value of 0.1 were considered statistically significant.
(TIF)

**S2 Fig.  Enriched pathways of the upregulated genes with ICR-R at old age.** Dot plot represents the enriched pathways of the upregulated genes in the ICR-R group compared to the AL group at old age. Adjusted p value of 0.1 were considered statistically significant.
(TIF)

**S3 Fig.  Enriched pathways of the downregulated genes with ICR-R at old age.** Dot plot represents the enriched pathways of the downregulated genes in the ICR-R group compared to the AL group at old age. Adjusted p value of 0.1 were considered statistically significant.
(TIF)

**S4 Fig.  Pathways affected by refeeding at old age.** Circular network plot displays the relationship between the DEGs and the enriched pathways in the ICR-RF group compared to the ICR-R group at old age, with fold-change values. Absolute log2 fold-change value of >2 and adjusted p value of 0.1 were considered statistically significant.
(TIF)

## Acknowledgments

The authors thank Dr. Margot P. Cleary (University of Minnesota, MN, USA) for generously donating the breeding colony of the MMTV-TGF-α transgenic mice. The authors thank both graduate and undergraduate students who helped with the daily feeding and handling of mice. The authors also thank Yeditepe University Animal Facility (YUDETEM) staff for taking care of the animals.

## Author contributions

**Conceptualization:** Bilge Guvenc Tuna, Soner Dogan, Sulev Koks.

**Data curation:** Bilge Guvenc Tuna, Nazim Arda Keles, Sulev Koks.

**Formal analysis:** Bilge Guvenc Tuna, Nazim Arda Keles, Munevver Burcu Cicekdal, Sulev Koks.

**Funding acquisition:** Sulev Koks.

**Investigation:** Bilge Guvenc Tuna, Soner Dogan, Sulev Koks.

**Methodology:** Bilge Guvenc Tuna, Nazim Arda Keles, Munevver Burcu Cicekdal, Soner Dogan, Sulev Koks.

**Project administration:** Soner Dogan.

**Resources:** Soner Dogan.

**Software:** Nazim Arda Keles.

**Supervision:** Bilge Guvenc Tuna, Munevver Burcu Cicekdal, Soner Dogan, Sulev Koks.

**Validation:** Nazim Arda Keles, Munevver Burcu Cicekdal, Soner Dogan, Sulev Koks.

**Visualization:** Nazim Arda Keles.

**Writing – original draft:** Bilge Guvenc Tuna, Nazim Arda Keles, Soner Dogan.

**Writing – review & editing:** Bilge Guvenc Tuna, Nazim Arda Keles, Munevver Burcu Cicekdal, Soner Dogan, Sulev Koks.

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
