## [Decision Letter · Decision Letter 0]

20 May 2025

Dear Dr. Koks,

Thank you for submitting your manuscript to PLOS ONE. After careful consideration, we feel that it has merit but does not fully meet PLOS ONE’s publication criteria as it currently stands. Therefore, we invite you to submit a revised version of the manuscript that addresses the points raised during the review process.

We look forward to receiving your revised manuscript.

Kind regards,

Osman El-Maarri, Ph.D

Academic Editor

PLOS ONE

Journal Requirements:

3. To comply with PLOS ONE submissions requirements, in your Methods section, please provide additional information regarding the experiments involving animals and ensure you have included details on (1) methods of sacrifice, (2) methods of anesthesia and/or analgesia, and (3) efforts to alleviate suffering.

This work was supported by Perron Institute for Neurological and Translational

Science, Nedlands, WA, Australia. This work was partly supported by The Scientific

and Technological Research Council of Turkiye (TUBITAK).

Reviewers' comments:

Reviewer's Responses to Questions

**Comments to the Author**

1. Is the manuscript technically sound, and do the data support the conclusions?

Reviewer #1: Yes

Reviewer #2: Yes

2. Has the statistical analysis been performed appropriately and rigorously?

Reviewer #1: Yes

Reviewer #2: Yes

3. Have the authors made all data underlying the findings in their manuscript fully available?

Reviewer #1: No

Reviewer #2: Yes

4. Is the manuscript presented in an intelligible fashion and written in standard English?

Reviewer #1: Yes

Reviewer #2: Yes

Reviewer #1: Your study clearly investigates the effects of caloric restriction (both chronic and intermittent) on the transcriptomic profile of mammary fat pad (MFP) tissue in a breast cancer mouse model, which is an important area in cancer research. It addresses the potential therapeutic effects of caloric restriction on age-related changes and tumor micro-environments.

Areas that probably need improvement: 1. you presented lots of data on gene expression changes, but they didn't explore the underlying mechanisms driving these changes. For example, the study identifies several upregulated genes, such as Malat1, Csn1s1, Csn2, but doesn't offer detailed mechanistic insights into how these genes influence cancer progression or age-related decline.

2. While the study is thorough, the findings are based on a single breast cancer mouse model. There are other breast cancer models, such as FVB/N-Tg(MMTVneu)202Mul/J andFVB/N-Tg(MMTV-Erbb2*,-cre)1Mul/J, that develops tumor without having to go through multiple rounds of pregnancies. You could cite other people's work using various breast cancer models and hypothesize whether your calories restriction method may or may not have an impact on other breast cancer models.

Experiments that can be done but not required: 1. validate the gene expression findings using qPCR of the most upregulated or downregulated genes, such as Malat1, Csn1s1, Cpt1b.

2. study the short-term re-feeding effects on gene expression in young mice following a period of intermittent caloric restriction, similar to the experiments done in old mice.

3. Perform knockdown or overexpression experiments for genes like Malat1 or Csn1s1 using siRNA in a mammary cell line to study their functions.

Reviewer #2: In this study, authors investigated how different caloric restriction (CR) protocols – chronic (CCR) and intermittent caloric restriction (ICR) – effect the whole transcriptome of mammary fat pad tissue (MFP) in a transgenic breast cancer mouse model (MMTV-TGF-α C57BL/6). They performed RNA sequencing of total RNA extracted from mice in different feeding groups (AL, CCR, ICR-R, and ICR-RF) at three life stages (baseline, adult, and old) and analyzed differential gene expression to explore age- and diet-associated transcriptomic changes. They showed that the long-term application of ICR but not CCR dramatically alters the whole transcriptome of the MFP tissue and re-feeding periods in ICR reverted the CR’s beneficial transcriptomic effects.

It is a well-designed study. Overall, the manuscript is well written and scientifically clear.

Minor comments:

- Line 35-37 “…then subjected to RNA sequencing” should be revised as “…were then subjected to..”

- Line 83 “These age-related molecular alterations associated with numerous diseases” should be revised as “…are associated with numerous diseases”.

- Line 362 “in the old age” – “at old age”.

- Line 370 “displasy” – “display”.

- Line 436 “breast cancer women” – “women with breast cancer”.

- Line 490 “Noteworthily” – “Notably”.

- The meaning of “DEGs” should be written in full when it is first mentioned in the text.

**Do you want your identity to be public for this peer review?** For information about this choice, including consent withdrawal, please see our Privacy Policy

Reviewer #1: **Yes: ** Huixin Wu

Reviewer #2: No

---

## [Author Response · Author response to Decision Letter 1]

17 Jul 2025

Dear Dr. Osman El-Maarri,

We, the authors, would like to thank the reviewers for their valuable and constructive comments, which have helped us to improve the quality and clarity of our manuscript. In response, we have carefully revised the manuscript in accordance with the reviewers’ suggestions. The changes are highlighted with track changes in the “Revised Manuscript with Track Changes” file. Our point by point response for each comments could also be seen below.

The revised version of the manuscript, revised manuscript with track changes, and the figures of the manuscript has been resubmitted via the editorial management system. We hope that the changes made will meet the expectations of the reviewers and the editor, and that the manuscript will now be considered suitable for publication.

Journal Requirements:

We have revised the file names and ensured that the manuscript adheres to all PLOS ONE’s style requirements, as outlined in the provided templates.

The ORCID iD of the corresponding author could not been authenticated in Editorial Manager due to a technical issue. However, we provide the ORCID iDs of all authors below:

Bilge Guvenc Tuna: 0000-0003-1348-1336

Nazim Arda Keles: 0000-0001-7118-1003

Munevver Burcu Cicekdal: 0000-0002-1986-0815

Soner Dogan: 0000-0002-7762-8109

Sulev Koks: 0000-0001-6087-6643

3. To comply with PLOS ONE submissions requirements, in your Methods section, please provide additional information regarding the experiments involving animals and ensure you have included details on (1) methods of sacrifice, (2) methods of anesthesia and/or analgesia, and (3) efforts to alleviate suffering.

We have revised the Methods section to include additional information on method of sacrifice, the use of anesthesia, and the efforts made to minimize suffering. These revisions have been included in the Section “2.1 Animals and experimental design” and are highlighted with track changes in the “Revised Manuscript with Track Changes” file.

This work was supported by Perron Institute for Neurological and Translational

Science, Nedlands, WA, Australia. This work was partly supported by The Scientific and Technological Research Council of Turkiye (TUBITAK).

The statement “The funders had no role in study design, data collection and analysis, decision to publish, or preparation of the manuscript.” has been added at the end of the “Funding” section in the revised manuscript.

The ethics statement has been moved to the Methods section, specifically at the end of section “2.1 Animals and experimental design” and removed from all other parts of the manuscript.

All references cited in the manuscript have been reviewed thoroughly. None of the articles cited in the manuscript have been retracted. The reference list has been formatted according to PLOS ONE’s reference style. The original article of the corresponding correction notice reference [3], has been added to the reference list [4]. Additionally, the reference [57], which was previously a “letter”, has been replaced with a suitable peer-reviewed reference.

Reviewer's Responses to Questions

Comments to the Author

3. Have the authors made all data underlying the findings in their manuscript fully available?

All data underlying the findings of this study have been made fully available. The raw sequencing reads have been deposited in the NCBI Sequence Read Archive (SRA) under BioProject ID PRJNA1279411.

Reviewer #1: Your study clearly investigates the effects of caloric restriction (both chronic and intermittent) on the transcriptomic profile of mammary fat pad (MFP) tissue in a breast cancer mouse model, which is an important area in cancer research. It addresses the potential therapeutic effects of caloric restriction on age-related changes and tumor micro-environments.

Areas that probably need improvement:

1. you presented lots of data on gene expression changes, but they didn't explore the underlying mechanisms driving these changes. For example, the study identifies several upregulated genes, such as Malat1, Csn1s1, Csn2, but doesn't offer detailed mechanistic insights into how these genes influence cancer progression or age-related decline.

2. While the study is thorough, the findings are based on a single breast cancer mouse model. There are other breast cancer models, such as FVB/N-Tg(MMTVneu)202Mul/J andFVB/N-Tg(MMTV-Erbb2*,-cre)1Mul/J, that develops tumor without having to go through multiple rounds of pregnancies. You could cite other people's work using various breast cancer models and hypothesize whether your calories restriction method may or may not have an impact on other breast cancer models.

We sincerely thank Reviewer #1 for the constructive feedback and valuable comments.

1. As suggested, we have expanded the Discussion section to include detailed mechanistic insights for key findings of the study. Specifically, we now discuss the potential roles of Malat1, Csn1s1, Csn2, Stxbp1, Snph, and Lep in breast cancer development and caloric restriction context, with new references. These changes could be seen with track changes in the “Revised Manuscript with Track Changes” file.

2. We acknowledge the limitation of using a single transgenic breast cancer mouse model. In the revised manuscript, relevant caloric restriction studies previously applied to other commonly used transgenic breast cancer mouse models are cited. We also discussed the differences in tumor latency and hormonal dependence, in the context of a response to caloric restriction. These modifications have been incorporated in the discussion part of the revised manuscript. These changes could be seen with track changes in the “Revised Manuscript with Track Changes” file.

Experiments that can be done but not required:

1. validate the gene expression findings using qPCR of the most upregulated or downregulated genes, such as Malat1, Csn1s1, Cpt1b.

2. study the short-term re-feeding effects on gene expression in young mice following a period of intermittent caloric restriction, similar to the experiments done in old mice.

3. Perform knockdown or overexpression experiments for genes like Malat1 or Csn1s1 using siRNA in a mammary cell line to study their functions.

We appreciate the valuable suggestions from Reviewer #1 regarding future experimantal directions insights to our study.

1. There is no question that this will add more mechanistic insights into the role of these genes in the early development of breast cancer and possible molecular background of the protective effect of caloric restriction. In our future studies, we plan to validate our RNA-seq findings with qPCR for the most up/downregulated genes, then we plan to proceed with the functional studies.

2. Our current study focused on the long-term effects of intermittent caloric restriction, and possible reverse effects of the refeeding period on the mammary fat pad transcriptome. We previously reported that the refeeding period of ICR did not effect the body weight, serum adiponectin and leptin levels at 14-week-old mice, but the weights of the mammary fat pad tissue was significantly different (Dogan et al., 2010, Oncology Letters). In this context, these findings suggest that MFP-specific effects of refeeding merit further investigation at the transcriptomic level in younger mice.

3. Following qPCR validations, we also plan to perform functional studies, such as knockdown or overexpression experiments in mammary epithelial cell lines, to further evaluate the mechanistic roles of these genes in cellular proliferation and early events in breast cancer development.

Reviewer #2: In this study, authors investigated how different caloric restriction (CR) protocols – chronic (CCR) and intermittent caloric restriction (ICR) – effect the whole transcriptome of mammary fat pad tissue (MFP) in a transgenic breast cancer mouse model (MMTV-TGF-α C57BL/6). They performed RNA sequencing of total RNA extracted from mice in different feeding groups (AL, CCR, ICR-R, and ICR-RF) at three life stages (baseline, adult, and old) and analyzed differential gene expression to explore age- and diet-associated transcriptomic changes. They showed that the long-term application of ICR but not CCR dramatically alters the whole transcriptome of the MFP tissue and re-feeding periods in ICR reverted the CR’s beneficial transcriptomic effects.

It is a well-designed study. Overall, the manuscript is well written and scientifically clear.

Minor comments:

- Line 35-37 “…then subjected to RNA sequencing” should be revised as “…were then subjected to..”

- Line 83 “These age-related molecular alterations associated with numerous diseases” should be revised as “…are associated with numerous diseases”.

- Line 362 “in the old age” – “at old age”.

- Line 370 “displasy” – “display”.

- Line 436 “breast cancer women” – “women with breast cancer”.

- Line 490 “Noteworthily” – “Notably”.

- The meaning of “DEGs” should be written in full when it is first mentioned in the text.

We sincerely thank Reviewer #2 for the valuable comments and corrections. All minor corrections have been addressed in the revised manuscript. Corrected parts were highlighted with track changes in the “Revised Manuscript with Track Changes” file.

While revising your submission, please upload your figure files to the Preflight Analysis and Conversion Engine (PACE) digital diagnostic tool, https://pacev2.apexcovantage.com/. PACE helps ensure that figures meet PLOS requirements. To use PACE, you must first register as a user. Registration is free. Then, login and navigate to the UPLOAD tab, where you will find detailed instructions on how to use the tool.

All figures have been uploaded to the PACE tool. The PACE-generated figure files have been downloaded and incorporated into the revised manuscript, in accordance with PLOS submission requirements.

Sincerely,

Sulev Koks, MD, Ph.D.

Head of Genetic Epidemiology Research

Perron Institute for Neurological and Translational Science

WA, Australia

---

## [Decision Letter · Decision Letter 1]

24 Aug 2025

Distinct transcriptomic effects of intermittent and chronic caloric restriction in mammary fat pad of a breast cancer mouse model

PONE-D-25-19116R1

Dear Dr. Koks,

We’re pleased to inform you that your manuscript has been judged scientifically suitable for publication and will be formally accepted for publication once it meets all outstanding technical requirements.

Kind regards,

Osman El-Maarri, Ph.D

Academic Editor

PLOS ONE

Additional Editor Comments (optional):

Reviewers' comments:

Reviewer's Responses to Questions

**Comments to the Author**

Reviewer #3: All comments have been addressed

2. Is the manuscript technically sound, and do the data support the conclusions?

Reviewer #3: Yes

3. Has the statistical analysis been performed appropriately and rigorously?

Reviewer #3: Yes

4. Have the authors made all data underlying the findings in their manuscript fully available?

Reviewer #3: Yes

5. Is the manuscript presented in an intelligible fashion and written in standard English?

Reviewer #3: Yes

Reviewer #3: (No Response)

**Do you want your identity to be public for this peer review?** For information about this choice, including consent withdrawal, please see our Privacy Policy

Reviewer #3: No
